



# Semiannual Variation in radiation belts particle fluxes: Van Allen probes observations

Facundo L. Poblet[1] and Francisco Azpilicueta[1]

[1]Facultad de Cs. Astronómicas y Geofísicas, Universidad Nacional de La Plata, La Plata, 1900, Argentina.

**Correspondence:** Facundo L. Poblet (fpoblet@fcaglp.unlp.edu.ar)

**Abstract.** The Semiannual Variation (SAV) is an annual pattern characterized by maxima around the equinoxes and minima near solstices observed in many space weather parameters. Several authors have studied this variation in the electron fluxes of the magnetosphere, focusing only in a few energy levels. In this investigation, Van Allen probes data are processed to extend SAV studies in electron fluxes of a wider energy range. A superposed epoch analysis was applied to data of the REPT and MagEIS instruments obtaining a clear semiannual pattern in the superposed year for L-shell values between 2.5 and 6.5. The Day Of Year (DOY) at which the highest electron flux values are detected near the September equinox coincide with the Russel & McPherron prediction. However, the DOY of the maximum expected close the March equinox occurs with a one month lag from the prediction of the accepted models. In addition, integrating over L-shell the annual DOY-L data with the semiannual pattern resulted in temporal curves that enabled to determine the energy range for which the SAV can be detected: from MeV to tens MeV energy values. Finally, an additional analysis of the fluxes of the Ring Current principal components ($H^+$ and $O^+$ ions) was performed, obtaining no evidence of a SAV on them. This result could indicate that the widely recognized semiannual pattern in the geomagnetic activity is related to a different current system.

## 1 Introduction

The tilted Earth's axis of rotation with respect to the orbital plane generates the seasons as the most recognizable characteristic while moving around the Sun. The differences in the exposure to the Solar radiation of the hemispheres along the year produces a seasonal variation in many weather parameters like the surface temperature or pressure. The geomagnetic Earth's dipole is also tilted with respect of the ecliptic giving rise to a similar, semiannual behavior in Space Weather parameters as well. Among these parameters are geomagnetic variables like the H component of the magnetic field at the surface of the Earth (Azpilicueta and Brunini, 2012) or the Sq focus location (Vichare et al., 2017), ionospheric parameters like vTEC (Azpilicueta and Brunini, 2011; Azpilicueta et al., 2012), and others. The Semiannual Variation (SAV) in all of them is characterized by maximum levels of activity near the equinoxes and minima near solstices.

One of the first mechanisms proposed to explain the SAV was reported by Cortie (1912) more than a hundred years ago. He realized that during the equinoxes the heliographic latitude of the Earth is close to its maximum absolute values, and suggested a connection between the high values of sun-spots number measured at these latitudes and the maxima detected in the magnetic activity. Actually, the Earth is exposed to regions where the solar wind velocity ($v_{sw}$) is high when the Earth goes through





these latitudes, which may be the driver of the enhancements in the activity. On the contrary, during solstices the Earth passes through regions of slow-speed solar wind near the Sun's equator and the activity is at a minimum (Phillips et al., 1995). This effect is known as the "axial" hypothesis. Later, Bartels (1932) identified a different angle as the possible controller of the SAV throughout the year: the colatitude of the subsolar point (the angle between the Earth's magnetic dipole and the Sun-Earth line).

The physical mechanism behind Bartels proposal was developed by Boller and Stolov (1970) demonstrating that the Kelvin-Helmholtz (KH) instability along the flanks of the magnetosphere exhibits a SAV (with instability maxima at the equinoxes and instability minima at the solstices). This is known as the "Equinoctial" hypothesis.

The last of the three mechanisms that are commonly referred in the literature to explain the SAV was introduced by Russell and McPherron (1973). The Russell & McPherron (RM) effect establishes that there is a varying probability of a southward

directed component of the Interplanetary Magnetic Field (IMF) in the Geocentric Solar Magnetospheric (GSM) coordinate system ($B_s$) throughout the year. This situation leads to different probability of magnetic reconnection between the IMF and the terrestrial magnetic field lines at the nose of the magnetopause. Near the equinoxes(solstices) the probability is maximum(minimum). As in the Axial and Equinoctial hypotheses, an angle controls the intensity of this mechanism during the year that is the angle between $z^{GSM}$ and $y^{GSEQ}$ (GSEQ: Geocentric Solar Equatorial coordinate system).

The RM hypothesis was developed after magnetic reconnection theories appeared (Petschek, 1964) and after the pioneer theoretical and experimental works made by J. W. Dungey in the 1960s (Dungey, 1961, 1963, 1965) revealed the importance of $B_s$ in generating geomagnetic activity. It was thought to be the natural answer to a long-standing problem. However, in the following years there was a considerable debate under the specific mechanism behind the SAV as well as in the significance of this variation (Cliver et al., 2000; Svalgaard et al., 2002; Svalgaard, 2011). Nowadays, the RM effect is the more often invoked

mechanism when studying the SAV in a specific parameter (Zhao and Zong, 2012; Bai et al., 2018).

In this work, the focus is on studying the SAV in the electron fluxes of the outer radiation belt with energies in the order of MeV. These fluxes show a strong SAV as has been found by several authors (Baker et al., 1999; Li et al., 2001; Kanekal et al., 2001). Baker et al. (1999) proposed that the driver mechanism behind it could be the result of a combination of the Equinoctial and the RM theories in the following manner. When high-speed solar wind streams encounter the Earth they can

cause substorms in case $B_s \neq 0$ as a consequence of the RM effect. The substorms then inject electron seed populations of low energy into the inner magnetosphere. Afterward these electrons are diffused inward by ULF waves in the Pc 5 band generated by the Kelvin Helmholtz instability, increasing their energy to relativistic values. On the other hand, several authors have attributed the major part of the SAV origin in the MeV fluxes to only one mechanism like Li et al. (2011) to the Equinoctial hypothesis or Kanekal et al. (2010) to the RM effect. Nevertheless, a unique consensus has not been reached yet.

To detect a SAV it is preferable (yet not mandatory) to have data of several years. This is the reason why many of the previous investigations of the SAV in electron fluxes has been performed with SAMPEX mission. The spacecraft was launched in 1992 and remained operative for more than one solar cycle. One of the motivations for this work is to extend these previous SAV studies utilizing Van-Allen probes (VAP) data. VAP spacecrafts have collected measurements that cover more than half of the 24th solar cycle so far and as it will be demonstrated, the results obtained in the following sections are comparable with the

previous investigations for the 23th (and part of the 22th) solar cycle .



The present study has been performed analyzing as many as energy levels VAP provides, with the objective of identifying the energy range in which the SAV is present. However, SAMPEX data were utilized as well, mainly for comparison purposes.

The remainder of this paper is organized as follows: Sect. 2 presents in detail the data, and their general characteristics in space and time are described in Sect. 3. The analysis of the semiannual pattern in the electron fluxes is developed throughout Sect. 4: from the processing scheme for obtaining a SAV to the analysis of the results.

In addition, the fluxes of the Ring Current principal components were processed to look for a SAV. This analysis is documented in Sect. 5. Finally, Sect. 6 presents the conclusions and implications of the work.

## 2  Data

This section describes the data that were utilized on this work. All the cited websites were publicly available at the completion of this paper around May 2018.

### 2.1  Van Allen probes particle fluxes

Van-Allen probes consists of two identical spacecrafts; RBSP-A and RBSP-B. Each spacecraft carries an instrument suite capable of measuring low-to-relativistic particle and electromagnetic environments in the Geospace. The twin VAP spacecrafts were launched in August 2012 and provide a unique opportunity to study a wide region that covers the inner and outer belts near the equator due to their highly elliptic orbits: $\sim 30500$ km in the apogee and $\sim 620$ km in the perigee, with only $10°$ of inclination. Electron flux measurements were taken from two instruments of the Energetic Particle Composition and Thermal Plasma Suite (ECT) (Spence et al., 2013): the Magnetic Electron Ion Spectrometer (MagEIS) (Blake et al., 2013) and the Relativistic Electron Proton Telescope (REPT) (Baker et al., 2013). Together, these instruments provide pitch angle resolved fluxes from 30 keV to 20 MeV electrons.

MagEIS electron data contain background contamination in some periods which is mainly caused by high-energy protons in the inner belt and in regions where multi-MeV electrons were present (Claudepierre et al., 2015). Therefore whenever was available, background-corrected electron flux data were used. However, this situation did not affect the results discussed below.

Section 5 used data from the Helium Oxygen Proton Electron mass spectrometer (HOPE) in the ECT suite which provides measurements of ion fluxes ($H^+$, $O^+$ and $He^+$) with energies between $\sim 1$ eV and 50 keV (Funsten et al., 2013). MagEIS also provides ion fluxes but without species discrimination. Therefore, it was assumed that MagEIS ion measurements correspond to protons only, as done by previous authors (Zhao et al., 2015).

On occasions, the results of a particular analysis will be attributed to a specific energy channel, for example 1.8 MeV REPT channel, but this actually means that the results correspond to a range of energies, in this example 1.6-2 MeV.

Finally, only RBSP-A data were employed in this study, downloaded for all the instruments from the OMNI website: ftp://spdf.gsfc.nasa.gov/pub/data/rbsp/.



## 2.2 SAMPEX electron Fluxes

Measurements from Solar, Anomalous, and Magnetospheric Particle Explorer (SAMPEX) were also utilized. SAMPEX was launched in July 1992 and orbited the Earth with a nearly circular orbit of $82°$ of inclination at a $600$ km altitude. The primary expressed goals of the SAMPEX program were to study (anomalous) galactic cosmic rays and solar energetic particle enhance-

ments. However it quickly became apparent the remarkable potential that SAMPEX had to study radiation belts processes due to the fact that its orbit examined all magnetic field lines from both Van Allen Belts (Baker and Blake, 2013). SAMPEX carried sensors capable of measuring high-energy electrons ($\gtrsim 1$ MeV) with the Proton-Electron Telescope (PET) (Cook et al., 1993) and the Heavy-Ion Large Telescope (HILT) (Klecker et al., 1993) but also had the capacity of measuring medium-energy electrons with HILT, and with other sensor, the low-energy ion composition analyzer (LICA) (Mason et al., 1993).

In this work, PET electron measurements were utilized, downloaded from the OMNI website ftp://spdf.gsfc.nasa.gov/pub/data/sampex/. The data from this website are calibrated fluxes that end up in June 2004 when the official SAMPEX NASA science mission ended. Averaged fluxes in two energy levels were considered: 3.75 and 8.25 MeV.

## 3    General Characteristics of the fluxes

Figure 1b-d show the complete electron fluxes data series in a YEAR vs L diagram. L is the geocentric distance at which

magnetic field lines would cross the magnetic equatorial plane, scaled in Earth radii. The energy channels correspond to 1.8, 3.8 and 5.2 MeV respectively. $\sim$60-90$°$ pitch angle are plotted (i.e. stably trapped particles) in logarithm scale. The different scales of the 3 energy channels located on the right indicate that the higher the energy, the lower are the fluxes values that the channel reachs. Particularly, the 1.8 MeV pannel presents the most intense and frequent enhancements.

The fluxes are limited between L $\simeq$ 3 and L $\simeq$ 6.5 that corresponds to the outer radiation belt. This belt is highly variable and

often shows electron intensities over 2 MeV. (Baker et al., 2016) realized that the outer belt encounter an almost impenetrable inner barrier at L $\simeq$ 2.8, practicaly no electrons with E $\geq$ 2.5 MeV are seen inside this region in the hours and days after major disturbances and their associated acceleration events and/or inward injections. There are in fact enhancements below L $\simeq$ 2.8 that occur in longer time scales associated with radial diffusion processes that could reach L = 2.5.

The internal barrier at L = 2.8 has important Space Weather consequences since any device moving from L = 2.5 to 2.8 can

expect almost complete protection from energetic electron fluxes. Note that below L $\simeq$ 2 there is an inner, more stable electron belt with an intensity that can be up to 5 orders of magnitude lower than the outer belt.

Figure 1a depicts the monthly averaged F10.7 index progression corresponding to the 24th 11-year Solar Cycle showing that VAP data cover the maximum activity period and the descending phase of this cycle. The maxima F10.7 values were reached in 2013-2014 and were particularly low compared with the maxima of previous cycles (Svalgaard, 2005). On the contrary, the

fluxes between the last part of 2013 until the beginning of 2015 were minimum, notably lower than other periods. Particularly for the 5.2 MeV panel the flux almost disappeared in mid 2014. The fluxes became maxima 2-3 years later in the descending phase of the cycle.





**Figure 1. (a)** Monthly averaged F10.7 index in sfu units (1 sfu = $10^{-22}$ W m$^{-2}$ Hz$^{-1}$). **(b-d)** Complete time series of ∼60-90° pitch angle fluxes from the ECT-REPT instrument on board the RBSP-A spacecraft. The fluxes for 1.8, 3.4 and 5.2 MeV are depicted.

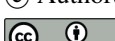



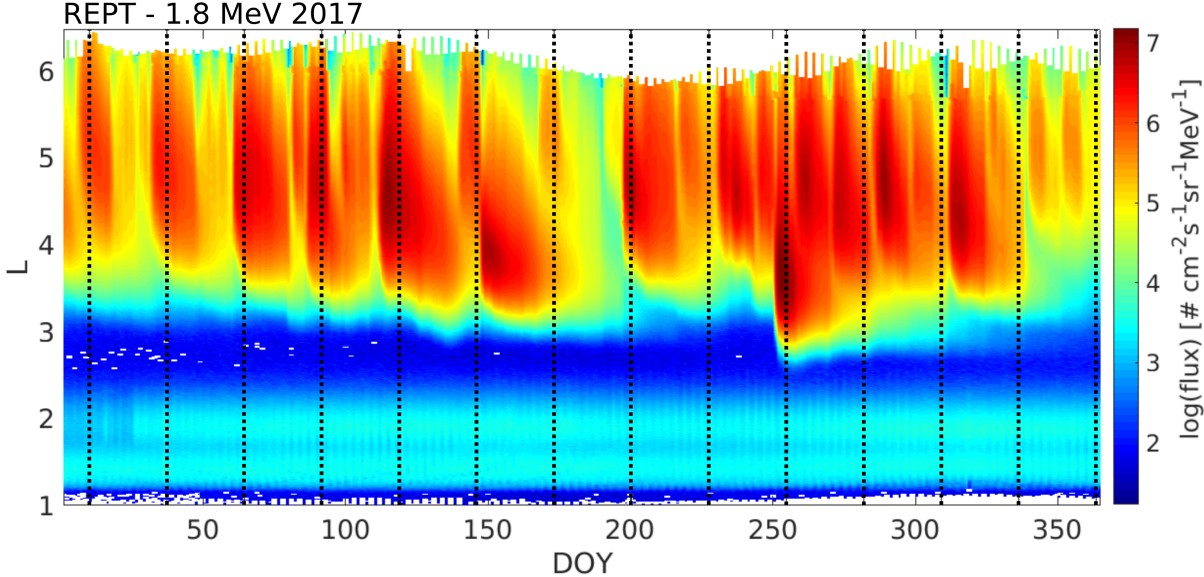

**Figure 2.** Electron flux measurements represented in a Day Of Year (DOY) vs L graphic for the year 2017 and 1.8 MeV. The data correspond to the ECT-REPT on board RBSPA spacecraft. The black dotted lines are separated 27.2 days between each other. Persistent enhancements and depletions over $\sim 27$ days are clearly seen.

The outer electron belt content has been associated with the solar wind, specifically with $\boldsymbol{v}_{sw}$, showing that when $v_{sw} \gtrsim 500$ $\mathrm{km\,s}^{-1}$ the electron content is high and when $v_{sw} \lesssim 300$ $\mathrm{km\,s}^{-1}$ persistently, outer belt electrons with $E \gtrsim 1$ MeV almost vanish (Baker et al., 2004).

In Fig. 2 the fluxes only for 1.8 MeV on 2017 are represented. The vertical black lines are separated by 27.2 days be-
5   tween them that corresponds to the synodic rotation period of the Sun. The most noticeable characteristics are the enhance-
ments(depletions) that arise repeatedly every 27 days approximately. Note also that in the equinoxes, the fluxes enhancements
penetrate deeper into lower L values.

During the descending phase of the solar cycle, high-speed streams emanating from long-lived coronal holes near the solar
equator has been found to last for a significant part of the solar cycle (Storini et al., 2006), generating a strong 27-day recurrence
10  in many solar-terrestrial parameters. However, this variation is not exclusively detected by the presence of coronal holes or any
specific structure but is the result of the longitudinal asymmetry of the Sun (Poblet and Azpilicueta, 2018).

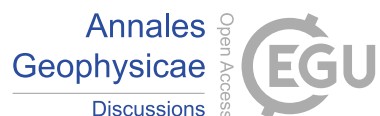



**Figure 3.** Superposed epoch analysis: **(a)** for REPT electron fluxes at 3.4 MeV and **(b)** for the SAMPEX PET electron fluxes. The vertical black lines are aligned with the DOYs of maximum activity that the three main hypotheses predict: Axial (dotted line), Equinoctial (dashed line) and RM effect (solid line).



**Table 1.** Dates at which the Axial, Equinoctial and RM effect predict maximum activity of the SAV. Adapted from (Cliver et al., 2002).

| Effect | First maximum | Second maximum |
|---|---|---|
| Axial | 7 March | 9 September |
| Equinoctial | 21 March | 23 September |
| Russell & McPherron | 7 April | 11 October |

## 4    Semiannual variation in the electron fluxes of the outer radiation belt

### 4.1    Superposed DOY-L characteristics

The SAV acts as a background intensity modulation of a specific phenomenon in the following manner. Suppose the solar wind encounter the Earth with values of $\boldsymbol{B}_{sw}$ and $v_{sw}$ at a random time of the year causing certain disturbances levels, then if the same solar wind values are detected near the equinoxes it is more probable to measure higher disturbances. Therefore, in order to obtain a Semiannual pattern from the electron fluxes, a superposition analysis in a pattern year was computed, calculating averages to diminish the effects of specific events and analyze only the SAV background modulation.

Averages of electron fluxes were calculated for each (Day Of Year (DOY) - L) bin using the complete data series (2012-2018). Then, a logarithm to the resulting superposition was calculated since the fluxes tend to increase and decrease several orders of magnitude many times during the year and finally a moving average with a centered window of 27 days wide was applied to diminish the 27-day variation that clearly appeared in Fig. 2 and could remain in the superposed year. The result of this procedure for the electron fluxes at E = 3.4 MeV can be observed in Fig. 3a. The three pairs of vertical lines indicate the DOYs for the maxima predicted by the three hypotheses mentioned in Sect. 1 i.e. the Axial (dotted line), Equinoctial (dashed line) and Russell & McPherron (solid line) hypotheses. These values are explicitly presented in Table 1.

Figure 3 shows maxima around the equinoxes of ~2 L wide centered at L = 4.2 that reflects a Semiannual pattern. The maximum near the March equinox has a double peak structure with its absolute maximum in DOY 126 and the maximum corresponding to the September equinox lies in DOY 285. Figure 3b presents the result of applying the same procedure to the SAMPEX data. As expected, the values of the two figures reach different intensities according the color scale located on the right (possibly due to they correspond to different time periods), yet the SAV pattern shows a similar behavior as well as the location of the maxima. The first maximum in Fig. 3b is in DOY 135 and the second one in DOY 288. They appear lower in L $\simeq$ 3.9.

The second maximum aligns almost exactly with the RM prediction for the SAMPEX and REPT data. However, slightly lower values align better with the Equinoctial prediction as well. Nevertheless, the striking characteristic of Figs. 3 (a) and (b) is that the first maximum is delayed almost one month from the RM prediction (and more from the other theories prediction).

Kanekal et al. (2010) noted this delay using 10 years of SAMPEX data, and suggested that the lack of the correct prediction of the first maximum location could be related to a lack of multiple solar cycles data, but the results presented here indicate that this is not related to the length of the observations but to an asymmetry in the physical process that energizes the electrons near



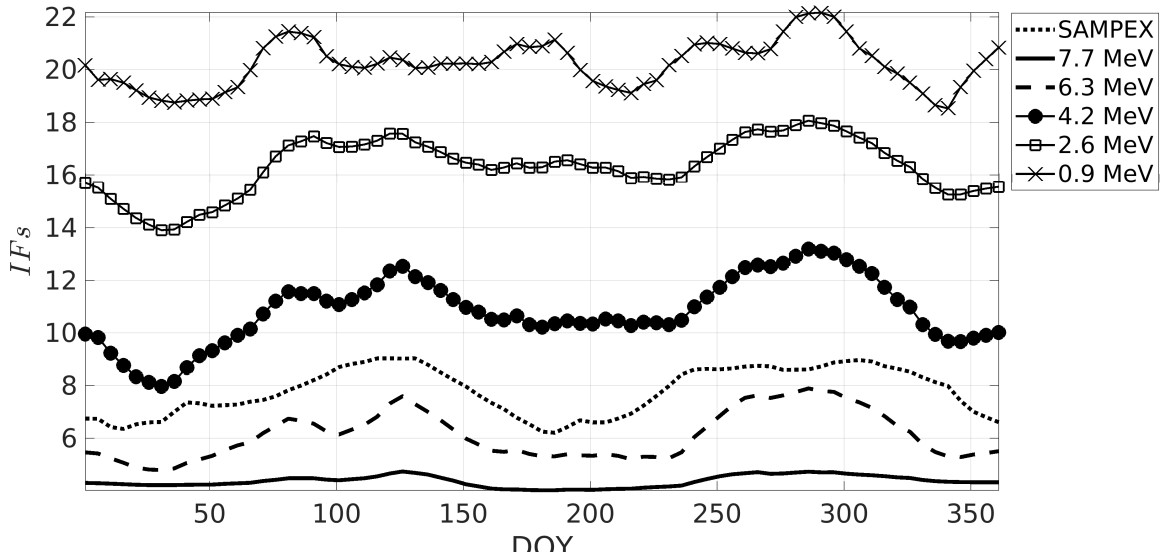

**Figure 4.** $IFs$ curves calculated with Eq. (1) for SAMPEX electron fluxes and for 5 energy channels of the REPT/MagEIS data: 0.9, 2.6, 4.2, 6.3 and 7.7 MeV.

the March and the September equinox respectively. Note that Fig. 3a shows high values aligned with Equinoctial prediction near the March equinox too.

## 4.2 Energy levels dependence of the SAV

In this section the SAV is analyzed in a broad energy range of the energetic electron fluxes. For this task REPT and MagEIS
5 offer a unique opportunity as was described earlier (Sect. 2).

From Fig. 1 and 3 it can be observed that the electron fluxes are located between L = 2.5 and L = 6.5, that are roughly the limits of the outer radiation belt. Hence an integration over L considering these limits was performed to the DOY-L Fluxes showed in Fig. 3. The following approximation was utilized:

$$IFs(\text{DOY}) = \Delta \text{L} \sum_{\text{L}=2.5}^{6.5} Fls(\text{DOY}, \text{L}), \tag{1}$$

where $Fls(\text{L}, \text{DOY})$ represents the fluxes in Fig. 3 and $IFs$ is the resulting curve in time. According to this procedure $IFs$
10 is a time sequence that integrates the components to the fluxes in every $L$ value of the outer radiation belt. $\Delta\text{L} = 0.1$ for the SAMPEX data and $0.01$ for the VAP data.

$IFs$ curves are shown in Fig. 3 for five different energy channels of the REPT/MagEIS data: 0.9, 2.6, 4.2, 6.3 and 7.7 MeV and for the SAMPEX data. Three main characteristics can be identified from this figure. The first one is that a semiannual behavior appears in all of them except in the 0.9 MeV curve. Note that the pattern becomes progressively less recognizable





**Table 2.** Intervals containing approximately three months of DOYs. They were elaborated so that the DOYs of the maxima predicted by the RM hypothesis (97 and 284) lie in the center of EQ1 and EQ2.

| Interval | DOYs (RM) |
|----------|-----------|
| EQ1 | 51-139 |
| SL1 | 140-231 |
| EQ2 | 232-323 |
| SL2 | 1-50, 324-365 |

as energy decreases. For the 4.2 and 5.2 MeV curves the SAV is very evident, the values located near the equinoxes are over one magnitude order higher than the values near solstices. $IFs$ at 7.7 MeV presents the semiannual pattern with the lowest equinoxes/solstices differences.

The second characteristic is that the annual mean value of each $IFs$ curve increases as the energies decrease, the maximum value corresponds to the 0.9 MeV curve and the minimum to the 7.7 MeV curve. This conclusion can also be reached examining the values in the color scale from Fig. 1. The third characteristic is that the maxima near the equinoxes and the minima near the solstices occur almost simultaneously for every energy channel. SAMPEX $IFs$ values present similar characteristics than the $IFs$ values for comparable energy levels of the VAP data.

In order to specifically evaluate the dependence of the Semiannual variation with the energy levels the following procedure was followed. The 365 values of each $IFs$ curve in the energy range delimited by 108 keV and 12.3 MeV were grouped into four intervals: EQ1, SL1, EQ2 and SL2 detailed in Table 2. Each interval comprises three months of $IFs$ daily values and they were arranged so that the DOYs of the maxima predicted by the RM hypothesis (97 and 284) lie in the center of EQ1 and EQ2 intervals. Note that each interval comprises approximately the days of the seasons, but with a delayed alignment with respect of the dates of the nominal equinoxes and solstices. Then, an averaged $IFs$ value was computed for each interval: $\overline{IFs}_i$, where $i = $ EQ1, SL1, EQ2, SL2; are the four intervals defined before.

Fig. 5a shows the result of calculating the following estimation for 5.2 MeV:

$$100 \left( \frac{\overline{IFs}_i}{\overline{IFs}} - 1 \right) \tag{2}$$

where $\overline{IFs}$ is the averaged $IFs$ value utilizing the 365 values of the year. Equation (2) represents the percentage deviation of each $\overline{IFs}_i$ from the yearly averaged value. The deviation for EQ1 is lower than EQ2 and the same occurs with SL intervals, the deviation from the mean of SL2 is higher than the deviation of SL1. Note that this occur almost invariably in all the curves of Fig. 4 (except in the SAMPEX curve). The maximum difference of ~31 % occur between EQ2 and SL2.

To estimate a mean difference between Equinoxes and Solstices, $\overline{IFs}_i$ values were averaged for EQ1 and EQ2 on one hand and for SL1 and SL2 on the other hand. The two values were then subtracted. The result for 5.2 MeV presented in Fig. 5a was ~21 %, lower than the ~31 % peak to peak deviation estimated before due to the calculation of the average between the two equinoxes and solstices intervals. This estimation was then extended to all the energy channels mentioned before (between 108 keV and 12.3 MeV) and the results are represented in Fig. 5b.





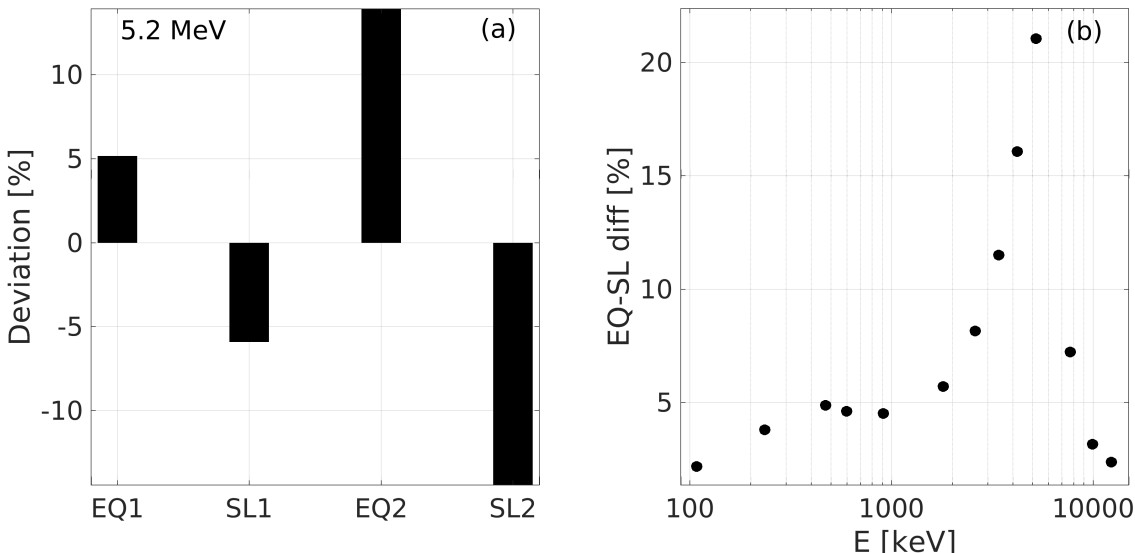

**Figure 5. (a)** Weighted deviations of the four $IFs_i$ seasonal values from the $IFs$ mean annual value for 5.2 MeV. **(b)** The values at EQ1 and EQ2 showed in (a) were averaged and then subtracted from the mean value calculated with SL1 and SL2 and plotted against their corresponding energy value (5.2 MeV). This calculation were extended for energy levels between 108 keV and 12.3 MeV.

From $\sim$100 keV to 909 keV (the last value in the keV range), the differences are below 5 %. Furthermore, the corresponding $IFs$ curve for 909 keV was depicted in Fig. 4 where a semiannual pattern can not be recognized. Hence, it can be inferred that the complete keV range does not present a SAV.

Moving to higher energies, the differences in the first energy channels with values in MeV start to grow, quickly reaching a maximum of $\sim$23 % in 6.3 MeV. Examples that these energies do present a recognizable semiannual pattern are the $IFs$ curves in Fig. 4 corresponding to 2.6, 4.2, 6.3 and 7.7 MeV. The differences remain above 5 % until the 7.7 MeV energy channel, then they fall quickly ending with a minimum in 12.3 MeV.

These results show that the SAV is confined only to a portion of the energy spectrum of the magnetospheric electron fluxes: from the MeV to tens MeV energy values.

## 5 Fluxes of the Ring Current components: SAV absence

The Ring Current (RC) is an electric current that surrounds the Earth moving westward. It is carried by a toroidal population of charged particles that in radial extension usually overlaps both, the cold dense plasma of the plasmasphere and the outer radiation belt.



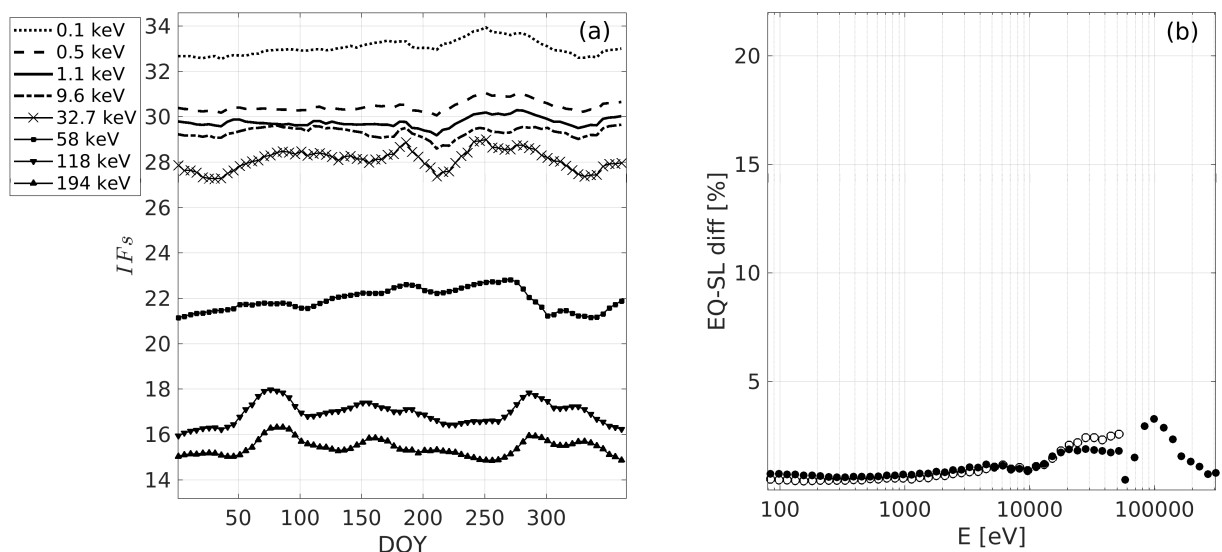

**Figure 6. (a)** $IFs$ curves calculated with Eq. (1) using HOPE and MagEIS proton fluxes of different energy levels: 0.1, 0.5, 1.1, 9.6, 32.7, 58, 118 and 194 keV. **(b)** The analogous calculation of the values presented in Fig. 5b but for protons in 0.08-308 keV and $O^+$ ions in 0.01-51.7 keV.

It is widely known that magnetic variations detected in the Earth's surface at low latitudes are strongly influenced by variations in the RC. In fact, specific indices has been developed to measure the geomagnetic response to the RC, the most famous is the Dst, and its modern version SYM-H (Wanliss and Showalter, 2006).

As it was mentioned in Sect. 1, the SAV has been long recognized in the Earth's magnetic activity but a link with a possible
5  SAV in the RC has not been reported. Hence, in this section the fluxes of the RC principal components are analyzed with VAP data in order to determine whether a SAV can be detected or not.

Ions in the range of tens keV to a few hundreds of keV has been found to be the main contributors of the total RC density (Williams, 1987; Zhao et al., 2016). Williams (1981) estimated that the bulk ($\sim$90 %) of the RC is contained in the 15-250 keV range and its composition primarily consists on $H^+$ and $O^+$ ions followed by a minor amount of other ionic species including
10  $He^{++}$ and $He^+$ (Hamilton et al., 1988; Daglis, 2007).

Figure 6a shows the $IFs$ curves calculated with Eq. (1) for HOPE and MagEIS proton fluxes for different energies between 0.1 keV and 194 keV. Unlike the curves of Fig. 4, a semiannual pattern can not be observed. Only the most energetic ones (118 keV and 194 keV) show modest peaks near the equinoxes but not enough to ensure a SAV. Moreover, the departure between the maximum and minimum value of every curve does not exceed $\sim$2 order of magnitude, lower than the maximum-minimum
15  differences of electron fluxes $IFs$ curves in Fig. 4.

The lack of a semiannual behavior in the main components of the RC is confirmed in Fig. 6b. This Figure presents the analogous of Fig. 5b where an estimation of the differences between $IFs$ values in the equinoxes and solstices were estimated,



but this time for protons and $O^+$ fluxes. The 0.08-308 keV range was covered but split in two: 0.08-51.7 keV using $O^+$ ions and protons of HOPE and 58-308 keV utilizing proton fluxes of MagEIS. The scale was maintained in order to facilitate the comparison with Fig. 5b. The differences are below 3.5 % for the entire energy range for both protons and $O^+$ ions.

## 6 Summary and conclusions

The results and analyses of the preceding sections have extended the knowledge of the SAV in magnetospheric particle fluxes. Among the novel findings is the determination of the energy range within which the SAV is detected in the energetic electron fluxes. The results show that the SAV is strictly confined between the MeV and tens MeV and that the characteristics are very similar in all the energy channels in this range. It is interesting to relate these results to the scenario proposed by Baker et al. (1999) mentioned in Sect. 1 in which the SAV could result from a combination of an Equinoctial and a RM effect. That proposal has two components. The first one are the electron fluxes in subrelativistic energies (tens to a couple of hundreds keV) that populate the Earth's magnetotail in response to a strong solar wind forcing event, typically a strong southward turning of the IMF. The second one are relativistic electrons ($\gtrsim 1$ MeV) that result from the energy amplification of the previously injected electrons due to the Boller-Stolov effect. According to the results of Sect. 4.2, the subrelativistic populations show little to none differences between equinoxes and solstices i.e. no SAV. Hence, the SAV seems to be present in only one of the two electron flux populations of the Baker et al. (1999) proposal.

It must be mentioned that the investigation of the mechanisms that accelerate electrons to relativistic levels in the magnetosphere is an active area of research. In fact, this was one of the scientific problems proposed to be solved with VAP data when the mission was designed.

The analysis of the location of the maximum near the September equinox suggests that the RM has a more prominent role in the SAV of relativistic electrons and the Axial effect the least important role. On the other hand, the situation is less clear for the maximum expected near the March equinox. The highest electron flux values occur almost one month after the RM prediction (the best estimation). The delay remains when a single year of observation is excluded from the data and also when a smaller number of years are superposed with VAP data. In the case of the SAMPEX data, the delay persists if the superposition is calculated with the years of the descending (1992-1996) and the ascending (1997-2001) phase separately. A complete solar cycle of VAP observations is not available yet to perform a similar test.

The previous authors suggestion indicating that this asymmetry in the maxima location corresponds to a short time span of the data might be discarded. The evidence presented here shows that the asymmetry is present in different solar cycles utilizing data from different missions. Hence, the phenomenon is not restricted to a particular solar cycle or data set. In any case, an explanation is still missing.

As well as in the electron fluxes in the keV range, the search for a SAV in the main components of the RC resulted in low differences between equinoxes intervals (EQ1 and EQ2) and the solstices intervals (SL1 and SL2), enabling to discard the presence of this variation in the RC. This result has consequences in the interpretation of the SAV detected in the geomagnetic activity measured in the ground, in the sense that it should come from a different current system other than the RC.



Finally, it must be pointed out that the results of this work are relevant from a Space Weather point of view. They show that a particular electron flux event could be 3 or 4 orders of magnitude more intense during the equinoxes than during solstices, and that the dates at which these increases and decreases occur vary with season. This semiannual modulation should definitely be considered in modeling of the radiation belts behavior.

5 *Data availability.* All the data utilized in this work can be freely downloaded from: ftp://spdf.gsfc.nasa.gov/pub/data/.

*Competing interests.* The authors declare that they have no conflict of interest.

*Acknowledgements.* The authors of this paper thank the Van Allen probes team for publicly distributing the data of REPT, MagEIS and HOPE through the OMNI website. The acknowledgments are extended to the SAMPEX team too.



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
