# Peer review of "Semiannual Variation in radiation belts particle fluxes: Van Allen probes observations"

_Annales Geophysicae, 2018_

## Referee Comment (RC1) · Anonymous Referee #1 · 6 Oct 2018

This manuscript describes a study of the semiannual variation in radiation belt and ring current data from the Van Allen Probes and SAMPEX missions. This is an interesting topic of study, and this manuscript may be suitable for publication after the authors address the comments below. The results could offer some interesting insight into the semiannual behavior of Earth's radiation belts and ring current. In particular, the results on the lack of a semiannual variation in the 10s of keV ring current ions and on the statistically stronger radiation belt response at the equinoxes are both good points of interest to the inner magnetosphere community. However, there are many major issues with the study that need to be resolved by the authors, and I cannot recommend this version of the manuscript/study for publication in Annales Geophysicae. The severity

of these issues should justify my rejection of this manuscript. I suggest that the authors address these issues with a revised and expanded study and resubmit a revised manuscript that details new results. These major issues are listed here:

1) It is troubling that the authors did not include any of the Van Allen Probes instrument PIs as coauthors on this study. Were the HOPE, MagEIS, and REPT PIs contacted for input on the results? The "Rules of the Road" for publication of Van Allen Probes data (see: https://www.rbsp-ect.lanl.gov/science/DataQualityCaveats.php) suggest that instrument PIs should be contacted for input and to validate the data being used for any study. The authors are not members of the Van Allen Probes science team, nor do they regularly participate in meetings attended by Van Allen Probes team members. Thus, I doubt they are experts in the data sets used or are aware of the various caveats in the data. If the authors had reached out to any of the Van Allen Probes science team, many of these issues listed in this review might have been avoided prior to this manuscript being submitted for review.

2) The RBSPICE instrument should be used in place of MagEIS for ions. Note the "major update A" on the rules of the road website listed above. There are considerable issues with MagEIS proton data. Thus, the validity of the results on the ring current ions here in the MagEIS range are questionable.

3) The results of this study do not span the electron energy range from "MeV to tens of MeV energy" as stated multiple times throughout this manuscript. The authors have apparently not considered the background levels of the instrumentation that they have used for the study, and the REPT has not measured ANY counts of >10 MeV above instrument background during the entire course of the mission. How the authors extrapolate from 7.7 MeV data to "10s of MeV" is not at all clear and actually quite dubious. Note that the only data of >10 MeV electrons shown in the entire manuscript are in Figure 5b, and that only shows up to ∼12 or 15 MeV. Thus, the results presented show nothing of "10s of MeV electrons". Furthermore, those three highest energy data points in that figure are likely entirely dominated by instrument background counts. No conclu-

sions can be drawn from those data, and it is misleading and inaccurate to show them at all. The effects of the instrument background in the statistics should be accounted for in all of the data used for this study.

4) Related to the last point, there have been very, very few instances of enhancements of the 6.3 and 7.7 MeV electron channels above background in REPT during the entire mission. If the authors included those channels in Figure 1, that would be immediately evident. Thus, they cannot conduct any reasonable statistical study with those data, since they are definitely dominated by only a few individual events. For this reason, I suggest the authors limit their study to <= 5.2 MeV considering this point.

5) The authors stress in the abstract and study description that they limited their study to 2.5 < L < 6.5. This is good since one of the major findings of the Van Allen Probes mission [e.g., Fennell et al., 2015, with MagEIS; X. Li et al., 2015, with REPT] is that the inner radiation belt is observationally devoid of any electrons with energy > 1.5 MeV. The REPT observations in that region are background contamination counts from penetrating protons (10s to 100s of MeV) in the inner belt. Those data there (at L < 2.5, e.g., Fig 1 and Fig 2) should not be shown; they are misleading and promote the outdated view that there are observable levels of multi-MeV electrons in the inner radiation belt. There are not. I strongly suggest the authors do not plot or study any data other than background corrected MagEIS electron data in the inner radiation belt. All L-shell plots with REPT data should be at L >= 2.5. The sentence on line 25-26 on page 4 ("Note that below L $\sim$ 2 there is an inner, more stable electron belt...") needs to be removed; it is not at all true. Those are protons, and the authors can see the Li et al. reference to support this.

6) The authors have not conducted an appropriate literature review of the most relevant results from the Van Allen Probes mission. They seem to have only read papers by Baker et al., based on their references. For example, the statement on line 26-27 of the Introduction ("Afterwards these electrons are diffused...") is outdated. Van Allen Probes results have shown that local acceleration is the dominant source of MeV

electrons in the outer radiation belts and provides the connection from 10s to 100s of keV electrons injected by substorm activity and the multi-MeV electrons in the outer belt. For examples supporting this see: Reeves et al. [Science 2013], Thorne et al. [Nature 2013], W. Li et al. [JGR 2014], Boyd et al. [GRL 2018] and many, many other papers from the Van Allen Probes mission. Also, in the summary and conclusions section, around line 11: what about the source of these electrons from substorm injections? See Meredith et al. [JGR 2003], Jaynes et al. [JGR 2015], and many references since that stress the importance of substorm activity for the source of these electrons. Their source is not just "in response to a strong solar wind forcing event" and the MeV electrons do not result from "the Boller-Stolov effect". The "Boller-Stolov effect" is considered relevant to radiation belt electrons, but it is most assuredly not considered the dominant acceleration mechanism for outer belt electrons. The authors are clearly not up to date with the latest radiation belt research. I suggest they conduct a more thorough literature review, particularly focusing on the latest results from the Van Allen Probes mission (including more than just Baker et al. papers) before they try to interpret the results of their study concerning radiation belt electrons.

7) There is a general lack of necessary detail in many of the figures to support the results of this study. For example: Why are no MagEIS data shown in Figure 1? Why are only 3.4 MeV data shown in Figure 3? Especially for the superposed epoch data (Fig 3) it is important to show examples from a wide range of relevant energies. Why aren't more energies shown for Figure 5a?

8) The results presented here for outer belt electrons are potentially heavily influenced by individual events due to the limited number of years used for this study. With only 5 years of data, there is technically only 5 "points" of averaged data in L and DOY space to include. For example, the March 2015 and September 2017 storms and low-L-shell enhancements of multi-MeV electrons are the only two events in which 3.4 MeV electrons were enhanced at L $\sim$ 3, especially considering the logarithmic nature of the flux enhancements (e.g, Figure 1 c), and these events likely strongly

affect the superposed results shown in Figure 3a! Those strong, geoeffective storms happened to fall around the equinoxes, likely because of the semiannual variation being studied here, but they are still just two individual events and nothing statistical can be claimed from two events that dominate the other four years. Those two events also dominate what is stated on lines 6-7 on page 6: "Note that in the equinoxes, the fluxes enhancements penetrate deeper into lower L-shells." . . . that is only because in those two events, which happened near the equinoxes there were very high levels of 1.8 MeV electrons. This entirely calls into question the validity of the results presented here.

9) The sentence on Lines 16-18 of section 3 (page 4) are unnecessary. Anyone familiar with studying radiation belt electrons knows that there are exponentially fewer electrons at higher energies than there are at lower energies. They typically follow power law or exponentially decaying distributions in energy. That's essentially all that this sentence is saying. . . there is nothing profound about the data showing this.

10) Nothing is mentioned of the peak in the electron fluxes at DOY ~85 or so in Figure 4. Why is that peak not relevant? It looks like it should be. Why are the three max activity lines from Figure 3 not also shown on Figure 4? They should be. Why aren't the peaks in the >100 keV ions around the equinoxes in Figure 6a considered relevant? How did the authors quantify what is a relevant variation and what is not? For example, what quantifies what is "enough" as stated on line 13 of page 12?

Other Minor and Typographical points:

1) The proper mission name is Van Allen Probes; note that this is the full title of the mission and "Probes" should be capitalized. This should be corrected throughout the manuscript.

2) The Van Allen Probes team and NASA Headquarters do not support or encourage the use of the acronym "VAP". Instead, the older acronym for the mission, RBSP (for Radiation Belt Storm Probes) is supposed to be used. VAP has negative connotations in the English language. I strongly encourage the authors to change all instances

of "VAP" in this manuscript to "RBSP". See the acronyms on the mission website: http://rbspgway.jhuapl.edu/ for support on this.

3) Is the "solar cycle" mentioned on line 9 on page 6 the 11-year solar cycle? This needs to be clarified. . .

4) On line 1 of page 6, "associated" should be "correlated"

5) What energy range is shown for SAMPEX in Figure 3?

6) What DOY are used for the start and end times (in 2012 and 2018, respectively) of the study?

7) Line 12 of page 9: "Fig. 3" should be "Fig. 4" here

8) The first and second paragraphs of section 5 (pages 11 and 12) are introduction material and belong better there.

―――――――――――――――――――――

---

## Referee Comment (RC2) · Anonymous Referee #2 · 8 Oct 2018

In their manuscript, 'Semiannual variation in radiation belts particle fluxes: Van Allen Probes observations', Poblet & Azpilicueta revisit the seasonal dependence of energetic and relativistic electron fluxes in the outer radiation belt using the Van Allen Probes data covering multiple energy channels. In particular, they analyse more than five years of data from the Van Allen Probes covering the period 2012-2018 to demonstrate electron flux peaks that are, however, not well aligned with equinoxes, similar to past results of Li et al. (2001) and Kanekal et al. (2010), from SAMPEX electron flux measurements. When they repeat the superposed epoch analysis with ring current ion flux measurements (for H+ and O+), a seasonal variation is not however clearly observed.

[Figure]

Though part of their study of energetic and relativistic electron fluxes using the new Van Allen Probes data set could complement and inform previous work, the manuscript is missing sufficient support for the statements made in the Summary and conclusions.

- Section 2 should be enriched with more explanation of the experimental data used for this study. Van Allen Probes have instruments designed to measure the relativistic electrons in both the inner and outer radiation belt. Radiation Belt Storm Probes (RBSP) was the name of the mission prior to launch, after which it was designated in honor of James Van Allen. The acronym RBSP is still used for the two spacecraft of the Van Allen Probes mission. The REPT experiment on board RBSP-A and RBSP-B has offered unprecedented observations of radiation belt electrons whose kinetic energy can reach up to 10 MeV - not up to 20 MeV [Baker et al. 2014, 2016] and an elaborate set of data products. For the MagEIS data set, a new release was announced on 25 May 2018 (please see https://www.rbsp-ect.lanl.gov/rbsp_ect.php) which has significant differences from the previous. Which data set have the authors used for this study, whether they analyse differential or integral flux, omnidirectional or directional flux and other essential information of the data set selected are not specified.

- In Section 3, the authors introduce what should be the McIlwain L-parameter or L-shell and not the dipole L. The model used for the calculation of the L-shell is not specified. Figure 1(b) – (d) suggest that an inner belt is seen at energies higher than 1 MeV which should be incorrect as the Van Allen Probes have revealed that the inner belt is largely devoid of relativistic electrons [e.g. Claudepierre et al., 2017]. Observations at L-shells lower than 2 should be omitted in Figure 1 and 2, as they have been in Figure 3. On page 6, lines 4-7, the description of Figure 2 needs to be revisited as it suggests Earthward swift of the outer belt around the autumn equinox while the spring equinox (DOY 79) is seen more than 40 days later. Electron flux variations are highly energy dependent in the outer radiation belt. How different is the electron flux variability seen at higher than 1.8 MeV energies?

- In Section 4, on page 8, the manuscript does not make clear whether the mean or

median of flux is shown in Figures 3 and 4. As noted by the authors, the electron fluxes vary over orders of magnitude especially during geomagnetically disturbed periods. During storms, relativistic electron fluxes are expected to abruptly drop and gradually increase more than three orders of magnitude as in the case of the January 1997 storm [Reeves et al. 1998]. As a result of the high variability in electron fluxes in the outer belt, the results of superposed epoch analysis should be characterised by significant spread around the median/mean. It would worthwhile to provide readers an indication of the spread, for example by choosing two different L-shells at which to quantitatively compare the distribution of electron flux values. In Figure 3, it should be the SAMPEX electron fluxes in the 1.5 MeV < E < 6 MeV energy range that are shown but this is specified. An alternative for the study of electron flux changes in the radiation belts in the total radiation belt electron content (Baker et al. 2004 included in the manuscript references list) which could remove reversible adiabatic changes observed during periods of geomagnetic activity and possibly improve the presented results.

- Without sufficient information about the data set used for the superposed epoch analysis of proton fluxes in the ring current, it is difficult to evaluate the suitability for the problem at hand or the validity of the results presented in Section 5. A combination of RBSPICE data for high energy protons and HOPE data for lower energy protons could be also be used. Communication with each experiment PI is necessary to get insights into the suitability of each data set as well as possible limitations that could have an effect on the analysis results. My concerns about the results regarding the ring current variability stem from the semiannual variation observed in the Dst index [e.g. Mursula & Karinen, 2005; Svalgaard, 2011] as well as the occurrence and intensity of storms [e.g. Temerin & Li, 2015], both of which are well established. And as the authors note on page 12, the Dst global 'ring-current' index is a measure of geomagnetic activity.

Previous studies on the electron radiation belts variability include:

Miyoshi et al. (2004), Solar cycle variations of electron radiation belts: Observations

and radial diffusion simulation, Space Weather, doi:10.1029/2004SW000070

Emery et al. (2011), Solar rotational periodicities and the semiannual variation in the solar wind, radiation belt and aurora, Solar Physics, doi:10.1007/s11207-011-9758-x

References not included in the manuscript list:

Baker et al. (2014), Gradual diffusion and punctuated phase space density enhancements of highly relativistic electrons: Van Allen Probes observations, GRL, doi:10.1002/2013GL058942

Claudepierre et al. (2017), The hidden dynamics of relativistic electrons (0.7–1.5 MeV) in the inner zone and slot region, JGR, doi:10.1002/2016JA023719

Mursula & Karinen (2005), Explaining and correcting the excessive semiannual variation in the Dst index, GRL, doi:10.1029/2005GL023132

Temerin & Li (2015), The Dst index underestimates the solar cycle variation of geomagnetic activity, JGR, doi:10.1002/2015JA021467

---

## Author Comment (AC1) · 14 Nov 2018

We are very grateful to the anonymous reviewer, it seems that she/he has a deep understanding of the subject of this manuscript.

We think that the two most significant corrections that could invalidate the results presented in our manuscript are the necessity of validate the quality of the data being processed and the possibility that the superposition analyses may be influenced by particular events considering the time span of the data.

With respect to the first point, we have already made contact with one PI of RBSP

mission and we are working in a new version of the manuscript. As far as the influence of particular events, we believe that it is not significant since our result are similar to the ones obtained by previous authors utilizing data from a different mission as is clearly stated in the manusctipt (e.g. analysis of Figure 3 and lines 19-29 of the Summary and conclusions). Moreover, we include a Figure in this response in which we exclude from the superposition two months of significant electron flux activity and the semiannual modulation can still be observed.

Below there is an answer to each comment of the reviewer. We hope we could submit an improved version of the manuscript to ANGEO.

This manuscript describes a study of the semiannual variation in radiation belt and ring current data from the Van Allen Probes and SAMPEX missions. This is an interesting topic of study, and this manuscript may be suitable for publication after the authors address the comments below. The results could offer some interesting insight into the semiannual behavior of Earth's radiation belts and ring current. In particular, the results on the lack of a semiannual variation in the 10s of keV ring current ions and on the statistically stronger radiation belt response at the equinoxes are both good points of interest to the inner magnetosphere community. However, there are many major issues with the study that need to be resolved by the authors, and I cannot recommend this version of the manuscript/study for publication in Annales Geophysicae. The severity of these issues should justify my rejection of this manuscript. I suggest that the authors address these issues with a revised and expanded study and resubmit a revised manuscript that details new results. These major issues are listed here:

1) It is troubling that the authors did not include any of the Van Allen Probes instrument PIs as coauthors on this study. Were the HOPE, MagEIS, and REPT PIs contacted for input on the results? The "Rules of the Road" for publication of Van Allen Probes data (see: https://www.rbsp-ect.lanl.gov/science/DataQualityCaveats.php) suggest that instrument PIs should be contacted for input and to validate the data being used for any study. The authors are not members of the Van Allen Probes science team, nor do they

regularly participate in meetings attended by Van Allen Probes team members. Thus, I doubt they are experts in the data sets used or are aware of the various caveats in the data. If the authors had reached out to any of the Van Allen Probes science team, many of these issues listed in this review might have been avoided prior to this manuscript being submitted for review.

ANSWER: We did not contact Van Allen Probes instrument PIs to process the data that appear in the manuscript. We avoided as much as we could the issues listed in the documents provided by the website mentioned by the reviewer. However, in these days we contacted one of the instruments PIs and we are working in a probable new version of the study.

2) The RBSPICE instrument should be used in place of MagEIS for ions. Note the "major update A" on the rules of the road website listed above. There are considerable issues with MagEIS proton data. Thus, the validity of the results on the ring current ions here in the MagEIS range are questionable.

ANSWER: The suggestion will be followed. The complete processing scheme will be repeated utilizing RBSPICE ion data and it will be compared with the results utilizing MagEIS, presented in the manuscript.

3) The results of this study do not span the electron energy range from "MeV to tens of MeV energy" as stated multiple times throughout this manuscript. The authors have apparently not considered the background levels of the instrumentation that they have used for the study, and the REPT has not measured ANY counts of >10 MeV above instrument background during the entire course of the mission. How the authors extrapolate from 7.7 MeV data to "10s of MeV" is not at all clear and actually quite dubious. Note that the only data of >10 MeV electrons shown in the entire manuscript are in Figure 5b, and that only shows up to âĹij12 or 15 MeV. Thus, the results presented show nothing of "10s of MeV electrons". Furthermore, those three highest energy data points in that figure are likely entirely dominated by instrument background counts.

No conclusions can be drawn from those data, and it is misleading and inaccurate to show them at all. The effects of the instrument background in the statistics should be accounted for in all of the data used for this study.

ANSWER: The comment is correct. The electron fluxes with the highest energy that were processed is 12.3 MeV. It is not accurate to write "tens of MeV". We will limit the analysis to lower energy channels in the future.

4) Related to the last point, there have been very, very few instances of enhancements of the 6.3 and 7.7 MeV electron channels above background in REPT during the entire mission. If the authors included those channels in Figure 1, that would be immediately evident. Thus, they cannot conduct any reasonable statistical study with those data, since they are definitely dominated by only a few individual events. For this reason, I suggest the authors limit their study to <= 5.2 MeV considering this point.

ANSWER: This suggestion will be followed in the future. However, processing these energy channels does not affect the main results presented in the manuscript and they serve as a complement in the analysis.

5) The authors stress in the abstract and study description that they limited their study to 2.5 < L < 6.5. This is good since one of the major findings of the Van Allen Probes mission [e.g., Fennell et al., 2015, with MagEIS; X. Li et al., 2015, with REPT] is that the inner radiation belt is observationally devoid of any electrons with energy > 1.5 MeV. The REPT observations in that region are background contamination counts from penetrating protons (10s to 100s of MeV) in the inner belt. Those data there (at L < 2.5, e.g., Fig 1 and Fig 2) should not be shown; they are misleading and promote the outdated view that there are observable levels of multi-MeV electrons in the inner radiation belt. There are not. I strongly suggest the authors do not plot or study any data other than background corrected MagEIS electron data in the inner radiation belt. All L-shell plots with REPT data should be at L >= 2.5. The sentence on line 25-26 on page 4 ("Note that below L áĹij 2 there is an inner, more stable electron belt. . .") needs

to be removed; it is not at all true. Those are protons, and the authors can see the Li et al. reference to support this.

ANSWER: Thank you for this comment. We were not aware of the Li et al. results mentioned.

6) The authors have not conducted an appropriate literature review of the most relevant results from the Van Allen Probes mission. They seem to have only read papers by Baker et al., based on their references. For example, the statement on line 26-27 of the Introduction ("Afterwards these electrons are diffused. . .") is outdated.

Van Allen Probes results have shown that local acceleration is the dominant source of MeV electrons in the outer radiation belts and provides the connection from 10s to 100s of keV electrons injected by substorm activity and the multi-MeV electrons in the outer belt. For examples supporting this see: Reeves et al. [Science 2013], Thorne et al.[Nature 2013], W. Li et al. [JGR 2014], Boyd et al. [GRL 2018] and many, many other papers from the Van Allen Probes mission.

Also, in the summary and conclusions section, around line 11: what about the source of these electrons from substorm injections? See Meredith et al. [JGR 2003], Jaynes et al. [JGR 2015], and many references since that stress the importance of substorm activity for the source of these electrons. Their source is not just "in response to a strong solar wind forcing event" and the MeV electrons do not result from "the Boller-Stolov effect". The "Boller-Stolov effect" is considered relevant to radiation belt electrons, but it is most assuredly not considered the dominant acceleration mechanism for outer belt electrons. The authors are clearly not up to date with the latest radiation belt research. I suggest they conduct a more thorough literature review, particularly focusing on the latest results from the Van Allen Probes mission (including more than just Baker et al. papers) before they try to interpret the results of their study concerning radiation belt electrons.

ANSWER: We are aware of some of the many discoveries and problems that Van Allen

Probes have helped to solve and we will incorporate a more extensive description of them regarding electron fluxes in another version of the manuscript. The Boller-Stolov effect is explained with more detail because is the only proven effect to have a Semiannual Variation. If the reviewer is aware of some other mechanism capable of accelerate electrons with a semiannual behavior we kindly suggest him/hem to mention it.

7) There is a general lack of necessary detail in many of the figures to support the results of this study. For example: Why are no MagEIS data shown in Figure 1? Why are only 3.4 MeV data shown in Figure 3? Especially for the superposed epoch data (Fig 3) it is important to show examples from a wide range of relevant energies. Why aren't more energies shown for Figure 5a?

ANSWER: For figures, in general we valued more the clarity of the argument that was intended to transmit than the fact that all the information be present. The superposition methodology used is technically easy to implement in case the reviewer wants to check additional curves. However, we will try to show more information in the figures or show them in the Appendix.

8) The results presented here for outer belt electrons are potentially heavily influenced by individual events due to the limited number of years used for this study. With only 5 years of data, there is technically only 5 "points" of averaged data in L and DOY space to include. For example, the March 2015 and September 2017 storms and low-L-shell enhancements of multi-MeV electrons are the only two events in which 3.4 MeV electrons were enhanced at L âĹij 3, especially considering the logarithmic nature of the flux enhancements (e.g, Figure 1 c), and these events likely strongly affect the superposed results shown in Figure 3a! Those strong, geoeffective storms happened to fall around the equinoxes, likely because of the semiannual variation being studied here, but they are still just two individual events and nothing statistical can be claimed from two events that dominate the other four years. Those two events also dominate what is stated on lines 6-7 on page 6: "Note that in the equinoxes, the fluxes

enhancements penetrate deeper into lower L-shells." . . . that is only because in those two events, which happened near the equinoxes there were very high levels of 1.8 MeV electrons. This entirely calls into question the validity of the results presented here.

ANSWER: From August 2012 (when RBSP was launched) until the first half of 2018 are almost 6 years, approximately a half of an 11-year solar cycle. The reviewer is right, there are still few points to make more robust statistical analyses. That is why we incorporated SAMPEX which has more than one 11-year solar cycle of data. The similarities that the SAV presents comparing SAMPEX data with RBSP data observed clearly in Figure 3 let us think that is not just coincidence. However, the reviewer did not mention any of this.

About the two enhancements on March 2015 and September 2017, we reproduced Figure 3 excluding the data of these two months and obtained the semiannual modulation (see Figure attached to this response) with essentially the same characteristics so results of the manuscript are not modified.

9) The sentence on Lines 16-18 of section 3 (page 4) are unnecessary. Anyone familiar with studying radiation belt electrons knows that there are exponentially fewer electrons at higher energies than there are at lower energies. They typically follow power law or exponentially decaying distributions in energy. That's essentially all that this sentence is saying. . . there is nothing profound about the data showing this.

ANSWER: The comment is correct. We will remove the sentence.

10) Nothing is mentioned of the peak in the electron fluxes at DOY âĹij85 or so in Figure 4. Why is that peak not relevant? It looks like it should be. Why are the three max activity lines from Figure 3 not also shown on Figure 4? They should be. Why aren't the peaks in the >100 keV ions around the equinoxes in Figure 6a considered relevant? How did the authors quantify what is a relevant variation and what is not? For example, what quantifies what is "enough" as stated on line 13 of page 12?

ANSWER: The three max activity lines will be incorporated in a future if similar figures are presented. The peaks around DOY ∼85 is mentioned in reference to Figure 3 (line 1 page 9). These peaks for the >100 keV curves in Figure 6a are dominant but when the mean differences between the equinoxes and solstices intervals (EQ1,2 an SL1,2 defined in the manuscript) are computed they result in low values, below 5% for all the ion energy channels, showed in Figure 6b. This means that there are not significant differences between the values in the equinoxes and the values in the solstices which is one of the main characteristic of the SAV. That is why these peaks are not considered relevant. On the other hand, the mean differences between the equinoxes and solstices intervals are above 20% for some MeV electron energy channels (Figure 5b).

Other Minor and Typographical points: 1) The proper mission name is Van Allen Probes; note that this is the full title of the mission and "Probes" should be capitalized. This should be corrected throughout the manuscript. 2) The Van Allen Probes team and NASA Headquarters do not support or encourage the use of the acronym "VAP". Instead, the older acronym for the mission, RBSP (for Radiation Belt Storm Probes) is supposed to be used. VAP has negative connotations in the English language. I strongly encourage the authors to change all instances of "VAP" in this manuscript to "RBSP". See the acronyms on the mission website: http://rbspgway.jhuapl.edu/ for support on this. 3) Is the "solar cycle" mentioned on line 9 on page 6 the 11-year solar cycle? This needs to be clarified. . . 4) On line 1 of page 6, "associated" should be "correlated" 5) What energy range is shown for SAMPEX in Figure 3? 6) What DOY are used for the start and end times (in 2012 and 2018, respectively) of the study? 7) Line 12 of page 9: "Fig. 3" should be "Fig. 4" here 8) The first and second paragraphs of section 5 (pages 11 and 12) are introduction material and belong better there.

[Figure]

**Fig. 1.** Fig. 3 of the manuscript.

[Figure]

**Fig. 2.** Fig. 3 of the manuscript without fluxes of March 2015 and September 2017.

---

## Author Comment (AC2) · 14 Nov 2018

We are very grateful to the anonymous reviewer. Beyond all the corrections, the reviewer has suggested many points that will significantly improve the quality of our work.

Below there is an answer to each comment of the reviewer. We hope we could submit an improved version of the manuscript to ANGEO.

In their manuscript, 'Semiannual variation in radiation belts particle fluxes: Van Allen Probes observations', Poblet & Azpilicueta revisit the seasonal dependence of energetic and relativistic electron fluxes in the outer radiation belt using the Van Allen

[Figure]

Probes data covering multiple energy channels. In particular, they analyse more than five years of data from the Van Allen Probes covering the period 2012-2018 to demonstrate electron flux peaks that are, however, not well aligned with equinoxes, similar to past results of Li et al. (2001) and Kanekal et al. (2010), from SAMPEX electron flux measurements. When they repeat the superposed epoch analysis with ring current ion flux measurements (for H+ and O+), a seasonal variation is not however clearly observed.

Though part of their study of energetic and relativistic electron fluxes using the new Van Allen Probes data set could complement and inform previous work, the manuscript is missing sufficient support for the statements made in the Summary and conclusions.

- Section 2 should be enriched with more explanation of the experimental data used for this study. Van Allen Probes have instruments designed to measure the relativistic electrons in both the inner and outer radiation belt. Radiation Belt Storm Probes (RBSP) was the name of the mission prior to launch, after which it was designated in honor of James Van Allen. The acronym RBSP is still used for the two spacecraft of the Van Allen Probes mission. The REPT experiment on board RBSP-A and RBSP-B has offered unprecedented observations of radiation belt electrons whose kinetic energy can reach up to 10 MeV - not up to 20 MeV [Baker et al. 2014, 2016] and an elaborate set of data products. For the MagEIS data set, a new release was announced on 25 May 2018 (please see https://www.rbsp-ect.lanl.gov/rbsp_ect.php) which has significant differences from the previous. Which data set have the authors used for this study, whether they analyse differential or integral flux, omnidirectional or directional flux and other essential information of the data set selected are not specified.

ANSWER: We utilized pitch angle resolved differential fluxes for RBSP data. The release of 25 May 2018 were processed for MagEIS data. We will expand this Section including more detail on the data employed.

- In Section 3, the authors introduce what should be the McIlwain L-parameter or L-shell

and not the dipole L. The model used for the calculation of the L-shell is not specified. Figure 1(b) – (d) suggest that an inner belt is seen at energies higher than 1 MeV which should be incorrect as the Van Allen Probes have revealed that the inner belt is largely devoid of relativistic electrons [e.g. Claudepierre et al., 2017]. Observations at L-shells lower than 2 should be omitted in Figure 1 and 2, as they have been in Figure 3. On page 6, lines 4-7, the description of Figure 2 needs to be revisited as it suggests Earthward swift of the outer belt around the autumn equinox while the spring equinox (DOY 79) is seen more than 40 days later. Electron flux variations are highly energy dependent in the outer radiation belt. How different is the electron flux variability seen at higher than 1.8 MeV energies?

ANSWER: The reviewer is correct. We used the McIlwain L-parameter calculated for 90 degree pitch angle particles with the OP77Q Olson & Pfitzer magnetospheric model. We were not aware of the results regarding the absence of relativistic electrons in the inner belt. We will correct Figures 1 and 2 to remove flux values below L = 2.5. Also the description of Figure 2 will be modified. We will add a similar analysis for higher energies.

- In Section 4, on page 8, the manuscript does not make clear whether the mean or median of flux is shown in Figures 3 and 4. As noted by the authors, the electron fluxes vary over orders of magnitude especially during geomagnetically disturbed periods. During storms, relativistic electron fluxes are expected to abruptly drop and gradually increase more than three orders of magnitude as in the case of the January 1997 storm [Reeves et al. 1998]. As a result of the high variability in electron fluxes in the outer belt, the results of superposed epoch analysis should be characterised by significant spread around the median/mean. It would worthwhile to provide readers an indication of the spread, for example by choosing two different L-shells at which to quantitatively compare the distribution of electron flux values. In Figure 3, it should be the SAMPEX electron fluxes in the 1.5 MeV < E < 6 MeV energy range that are shown but this is specified. An alternative for the study of electron flux changes in the

radiation belts in the total radiation belt electron content (Baker et al. 2004 included in the manuscript references list) which could remove reversible adiabatic changes observed during periods of geomagnetic activity and possibly improve the presented results.

ANSWER: Thanks for this comment. The many suggestions could significantly improve the manuscript. In Figure 3 and 4, mean values were calculated.

- Without sufficient information about the data set used for the superposed epoch analysis of proton fluxes in the ring current, it is difficult to evaluate the suitability for the problem at hand or the validity of the results presented in Section 5. A combination of RBSPICE data for high energy protons and HOPE data for lower energy protons could be also be used. Communication with each experiment PI is necessary to get insights into the suitability of each data set as well as possible limitations that could have an effect on the analysis results. My concerns about the results regarding the ring current variability stem from the semiannual variation observed in the Dst index [e.g. Mursula & Karinen, 2005; Svalgaard, 2011] as well as the occurrence and intensity of storms [e.g. Temerin & Li, 2015], both of which are well established. And as the authors note on page 12, the Dst global 'ring-current' index is a measure of geomagnetic activity.

ANSWER: We have already made contact with an ECT PI for REPT and MagEIS data and we will do the same for RBSPICE data. As the reviewer suggests, a combination of RBSPICE and MagEIS data could change our results of the apparent absence of a semiannual pattern in the fluxes of the ring current principal components which is very well known in the geomagnetic activity.

Previous studies on the electron radiation belts variability include:

Miyoshi et al. (2004), Solar cycle variations of electron radiation belts: Observations and radial diffusion simulation, Space Weather, doi:10.1029/2004SW000070 Emery et al. (2011), Solar rotational periodicities and the semiannual variation in the solar wind, radiation belt and aurora, Solar Physics, doi:10.1007/s11207-011-9758-x

References not included in the manuscript list:

Baker et al. (2014), Gradual diffusion and punctuated phase space density enhancements of highly relativistic electrons: Van Allen Probes observations, GRL, doi:10.1002/2013GL058942 Claudepierre et al. (2017), The hidden dynamics of relativistic electrons (0.7–1.5 MeV) in the inner zone and slot region, JGR, doi:10.1002/2016JA023719 Mursula & Karinen (2005), Explaining and correcting the excessive semiannual variation in the Dst index, GRL, doi:10.1029/2005GL023132 Temerin & Li (2015), The Dst index underestimates the solar cycle variation of geomagnetic activity, JGR, doi:10.1002/2015JA021467